# Efficient Compressed Sensing for Real-Time Electrocardiogram Acquisition on Low-Power Medical Devices

Mitchell Terrell
*Dept. of Computer Science and Engineering*
*University of Minnesota*
Minneapolis, Minnesota
terre101@umn.edu

Jon Weissman
*Dept. of Computer Science and Engineering*
*University of Minnesota*
Minneapolis, Minnesota
weiss039@umn.edu

*Abstract*—Low cost wearable and implantable cardiac monitoring devices (WCM & ICM), combined with increasingly accurate disease and arrhythmia detection algorithms have proven effective to help slow the impact of cardiovascular disease, the worlds leading cause of death in 2022 [1]. Improvements to server-side detection algorithms along with hardware limitations of these devices such as slow processor speeds and minimal battery life has led to a desire to offload data from the devices for later analysis. Paired with limited storage capacity, this has led to a push to decrease the necessary storage for electrocardiogram (EKG) signals without sacrificing disease detection accuracy and device longevity. A promising recent innovation in EKG compression has come from Compressed Sensing (CS), which exposes inherent sparseness in the signal to selectively sample for eventual server-side reconstruction. Many CS approaches have been implemented on WCM devices which demonstrate a high compression ratio (CR) and accurate signal reconstruction, but quickly become impractical for the stricter hardware constraints of ICM devices due to the increased computation on the device and slow reconstruction time. In this paper we propose a CS approach known as Tailored Sensing (TS) which combines on-device prior knowledge, a custom transform basis, and optimal sense location selection to achieve improved CR and reconstruction accuracy while eliminating on-device computational burden and slow reconstruction time. Our approach offers equivalent or better signal reconstruction at previously unfathomable CRs due to our novel signal segmentation scheme. Additionally, our approach boasts a zero overhead on-device sensing strategy and a 93% reduction in signal reconstruction time.

*Index Terms*—compressed sensing, electrocardiogram, tailored sensing, signal reconstruction, implantable medical devices

## I. Introduction

With the onset of *big data*, *machine learning*, and *artificial intelligence*, no industry is immune to the changes necessary to adapt to this new era. The medical device industry is no exception to this fact. Wearable medical device technology has already begun to adjust to this change with easily updated and continually improving detection algorithms. On the other hand, implanted devices have been slow (or unable) to capitalize on these advancements. To the benefit of the patient, ICM devices must go through intense regulatory bodies such as the Food and Drug Administration and the European Medicines Agency before a device can ever reach a patient. With this regulation comes slower, more deliberate innovation, but promotes patient safety. Despite this fact ICM devices still remain the defacto standard for cardiac electrophysiologists because of the increased signal accuracy that comes with being implanted directly next to the heart. In pursuit of improved arrhythmia detection and analysis of cardiac degradation overtime, many on-device algorithms are implemented, but suffer from an inability to be adjusted once implemented and minimal knowledge of change over time. The combination of the aforementioned factors has presented a demand to offload the highly accurate data that the implanted device is sensing and analyze it externally for diseases and arrhythmias. This would circumvent the strict regulation that comes with modifying the firmware on an existing device and allow for easily deployed detection algorithm improvements.

The ability to offload data from an ICM device for server side analysis is almost entirely infeasible at this time. This is largely due to the memory constraints on these tiny devices. For example, a single sense vector for voltage difference across the heart, sampled at a rate of 300 Hz with 12 bits of resolution generates almost 39 MB of data in a single day. Considering that these devices can be smaller then a human thumb and have many other sensors on them (accelerometer, minute ventilation, temperature) one can see how storing all of this information can quickly become infeasible. Regardless of whether an entire days worth of signal could be stored on a device, methods such as remote patient monitoring for heart failure [2] and Atrial Fibrillation detection [3] have shown that even small segments of EKG data collected daily and offloaded periodically could be used to extract meaningful information.

In this paper, a novel CS approach is proposed as a solution to reduce the amount of memory necessary to store and transmit EKG signals to help make server-side detection algorithms feasible for ICM devices. Modern telemetry technology such as Bluetooth® paired with the increasingly popular bedside monitors and mobile applications for data offloading, can allow server based EKG analysis to become a reality.

**TABLE I**
**SYMBOL TABLE**

| Symbol | Description |
|---|---|
| $N$ | Number of Uncompressed Samples |
| $M$ | Number of Compressed Samples |
| $K$ | Sparsity of $x$ in $\Psi$ |
| $x \in \mathbb{R}^N$ | Uncompressed EKG Signal |
| $y \in \mathbb{R}^M$ | Compressed EKG Signal |
| $s \in \mathbb{R}^N$ | Universal Basis Coefficients |
| $a \in \mathbb{R}^M$ | Tailored Basis Coefficients |
| $C \in \mathbb{R}^{M \times N}$ | Measurement Matrix |
| $\Psi \in \mathbb{R}^{N \times N}$ | Universal Transform Basis |
| $\Psi_r \in \mathbb{R}^{N \times M}$ | Tailored Transform Basis |
| $\Theta = C \times \Psi$ | Product of Measurement and Basis |

*Adapted from [10]

## A. Related Work

Compressed sensing of EKG signals is a well researched subject with many existing CS approaches both proposed in literature and deployed in applications. Each of these approaches offer improvements to one or many of the core aspects of CS such as compression ratio, signal reconstruction accuracy, reconstruction time, or on-device computation. The existing state-of-the-art CS methods often fail to take into account a holistic practicality of the approach, focusing on improving one aspect of CS such as reconstruction accuracy at the expense of another CS aspect such as lengthy reconstruction times.

Many of these CS approaches present optimizations to the signal transform basis ($\Psi$) for generic signal acquisition and reconstruction. EKG signals have been shown to be optimally sparse in discrete cosine transform (DCT) basis [4], [5] as well as the discrete wavelet transform [6], [7] allowing for generic sampling of the EKG signal, but these methods suffer from lower compression ratios and higher reconstruction errors because of the number of samples required to satisfy the incoherence restrictions between the transform basis ($\Psi$) and the measurement matrix ($C$). Other CS methods aim to optimize $C$ to reduce the number of necessary samples or optimize the sampling locations. Measurement matrices such as a Gaussian normal distribution [5] offer decent signal reconstruction accuracy, but require an on-device matrix multiplication to be performed for each compression window. More practical binary sampling matrices such as Bernoulli random matrices [8] or Deterministic Binary Block Diagonal (DDBD) [6], [9] matrices offer multiplication-less measurement, but still require summation operations that scale with the signal density causing unnecessary battery drain on the device.

A learned over-complete dictionary of EKG signals was used to develop a non-generic transform basis $\Psi$ which was used in combination with a DDBD based $C$ [6] to offer nearly equivalent signal reconstruction accuracy to the proposed method at equivalent CRs. Although equally as accurate as the proposed method, the DDBD based measurement matrix lead to a larger computational impact on the device and still required expensive sparse optimization to reconstruct the signal.

## II. BACKGROUND

### A. Compressed Sensing

The theory behind compressed sensing is rooted in the fact that natural signals are inherently sparse in the appropriate coordinate system making them highly compressible when represented by a sparse vector of coefficients. Sparsity in natural signals, specifically EKG signals, is best illustrated by understanding the vastness of EKG signal space. To put this into perspective, consider a 1 second EKG sample with 12 bit resolution sampled at 300 Hz. There are more than $2^{3600}$ unique signals that can be derived from this signal space, which exceeds the number of atoms in the known universe. Selecting a single natural EKG signal from that signal space demonstrates the sparsity of natural EKG signals within EKG signal space.

*1) Mathematical Formulation for Compressed Sensing::*
The uncompressed EKG signal can be represented in it's original state as a high dimensional vector $x \in \mathbb{R}^N$ and is known to be $K$ sparse in a generic transform basis $\Psi \in \mathbb{R}^{N \times N}$ implying that $x$ can be represented in the basis of $\Psi$ with $K$ non-zero elements, exposing a sparse representation of $x$:

$$x = \Psi s \qquad (s \in \mathbb{R}^N) \tag{1}$$

A compressed measurement of $x$ can be represented by $y \in \mathbb{R}^M$ where $K < M << N$. The measurement matrix $C \in \mathbb{R}^{M \times N}$ is a set of $M$ measurements of $x$ where $C$ is typically made up of random measurements of $x$ such that $C$ is *incoherent* with $\Psi$ and the number of compressed samples $M$ are sufficiently large ($K < M$). A sparse vector $s \in \mathbb{R}^N$ is able to reconstruct the original high dimensional signal in Eq. 2.

$$y = C\Psi s = \Theta s \tag{2}$$

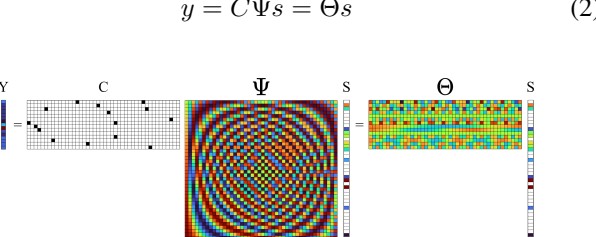

Fig. 1. Compressed sensing of an EKG signal

This is shown visually in Fig. 1 with the compressed sample of an EKG signal $y$ represented with a sparse random measurement matrix $C$ and a DCT basis $\Psi$.

An undetermined system of equations is represented in Eq. 2 whose solution is derived by the sparsest representation of $s$, defined by the minimization of the $l_0$ norm of $s$ ($\|s\|_0$) which is a combinatorial hard (NP-hard) problem to solve. Fortunately, recent advances in applied mathematics have shown that under certain conditions of $C$ the constraint on $s$ can be relaxed to a convex $l_1$ norm optimization [11] shown in Eq. (3).

$$\hat{s} = \arg\min_s \|s\|_1 \tag{3}$$

Where $\hat{s}$ can be solved for with sparse optimization techniques such as OMP, IRLS and BP such that $y = C\Psi\hat{s}$ Therefore a compressed measurement $y$ with a known measurement matrix $C$ and transform basis $\Psi$ can be used to determine a sparse vector of coefficients $\hat{s}$ which can then be used to reconstruct the original signal into $\hat{x}$: $\hat{x} = \Psi\hat{s}$

### B. Tailored Sensing

Compressed sensing is based on signal reconstruction in a generic basis such as a DCT basis or a wavelet basis which works well if there is no prior knowledge of the signal. On the other hand, tailored sensing suggests that an even sparser representation may exist for a given signal if the general structure of the signal is known prior to sampling. TS proposes that a signal is optimally sparse in a basis formed from a library of features built from the decomposition of that signal via the singular value decomposition (SVD) or proper orthogonal decomposition (POD). It is thus proposed that optimal sense locations may exist for a signal represented in this tailored basis.

*1) Mathematical Formulation for Tailored Sensing::* TS is based on the formation of a tailored library $\Psi_r \in \mathbb{R}^{N \times M}$ from the decomposition of existing examples of a particular signal such that $y = C\Psi_r a = \Theta a$ Where the measurement matrix $C$ is constructed such that $\Theta$ can be inverted to identify a low rank vector of coefficients $a \in \mathbb{R}^M$ represented visually in Fig. 2.

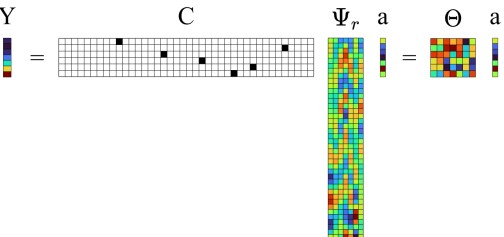

Fig. 2. Tailored sensing of an EKG signal

The goal of tailored sensing is to design $C$ to minimize the condition number of $\Theta$. The condition number of $\Theta$ is a measure of the maximum and minimum singular values which proves to be indicative of how sensitive $\Theta$ is to matrix multiplication or inversion [11]. The sensitivity of $\Theta$ maps directly to the signal-to-noise ratio (SNR) of a reconstructed signal sampled by the measurement matrix $C$ and represented in the tailored basis $\Psi_r$. In simpler terms, sampling the locations of the signal $x$ which activate the most significant features of the tailored basis $\Psi_r$ will allow for the reconstruction of the signal with an optimally low SNR.

The selection of these optimal sampling locations has been shown to be easily obtained using greedy optimization procedures such as QR matrix factorization [10]. QR factorization utilizes column pivoting to decompose a given matrix $B$ into an upper triangular matrix $R$, a unitary matrix $Q$, and a column permutation matrix $C^T$. This can be represented as $BC^T = QR$

The QR factorization of the tailored basis $\Psi$ can thus yield optimal sampling locations by utilizing the pivot locations from the column permutation matrix $C^T$ to construct the measurement matrix $C$ as the pivot locations of tailored basis represent the optimal sample of $r$ basis modes of $\Psi$ leading to $\Psi_r$. This can be represented as $\Psi_r^T C^T = QR$.

Putting this all together, a compressed measurement of the original signal $x$ can be formed using the pivot locations from the QR factorization $C^T$. The compressed measurement $y$ can then be used directly to solve for low-rank vector of coefficients $a$ by multiplying by the pseudo-inverse of $\Theta$ shown as $\Theta^+$. Finally, $a$ can then be used to reconstruct the original signal into $\hat{x}$ thus $\hat{x} = \Psi_r a$

## III. PROPOSED METHOD

This paper proposes a novel TS based solution for real-time EKG data acquisition designed for use in lower power medical devices. Existing CS approaches assume no prior knowledge of an EKG signal's variance in time. This means for a continuous EKG signal these CS methods have a limit to how much they can down-sample the signal while maintaining accurate reconstruction since they cannot make any assumptions on the number of heart beats that might occur in a given window of compression. Furthermore other TS methods which employ a tailored basis for the signal [6] must sample a sufficiently large percentage of the signal to account for time shifts in the key features of the signal (i.e. the location of the beat).

Due to the hardware limitations of implanted medical devices these devices rely heavily on analyzing the EKG signal in real time and only saving the most pertinent information. Additionally, devices that are required to deliver therapy, such as high voltage defibrillation, must be able to accurately characterize the behavior of the heart in real-time. Because of these requirements, implanted medical devices already contain built in EKG analysis tools such as beat detection, heart rate tracking and signal noise detection.

The method proposed in this paper utilizes the built in beat detection on these devices to slice the EKG signal into compressible single beat windows which are optimally sparse in a tailored basis and have known sample locations which best characterize the features of the beat.

### A. Signal Segmentation & Sampling Scheme

Implanted medical devices require advanced on-device signal analysis in order to properly treat a patient as well as to detect when abnormal behavior is occurring in the heart for episode storage. One of the most important of these on-device analysis tools is EKG peak detection. The peak of an EKG signal is known as the R-wave and is the primary indicator of a heart beat, thus it is very important for these devices to be able to accurately detect the location of these R-waves. The method proposed in this paper exploits this functionality to slice an uncompressed EKG-signal into compressible single-beat windows.

As a continuous EKG signal is being collected on a device, the location of the peak is determined in digital hardware and

provided to the firmware in real-time. The firmware is then able to sample this beat at optimal locations determined from a distance in time of the location of the peak. A pictorial example of the segmented beat windows plotted on top of each other is provided in Fig. 3 where the top figure shows the original beats and the bottom shows the same beats normalized for use in the tailored basis generation.

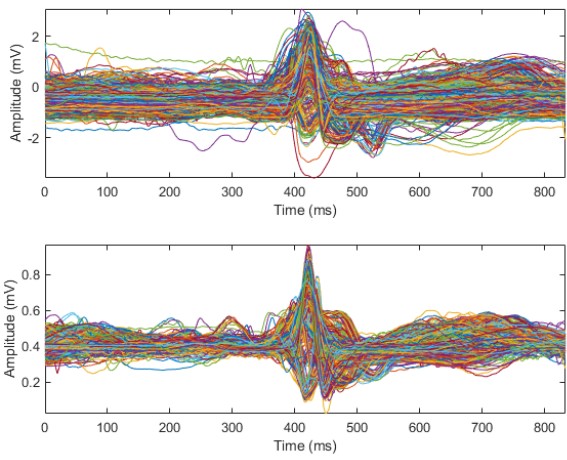

Fig. 3. Original (top) and normalized (bottom) EKG windows sliced and overlaid.

This eliminates the need for generality in our compressed sensing scheme, since there is no need to account for the possibility of multiple beats nor shifts in the beat location within the sampling window. Additionally, this does not create any additional computation overhead on the device since beat detection is built into cardiac monitoring devices.

### B. Tailored Basis Generation

A given natural signal has been shown to be optimally sparse in a basis formed from the decomposition of the signals own features into what is known as a tailored basis. This tailored basis can be formed from the decomposition of a library of these signals via the SVD, POD or other matrix factorization methods.

Since our method utilizes single-beat windows for compression, a tailored basis $Psi_r$ was created based on single-beats. Tailored bases are built from the decomposition of an over-complete library of a given signal; to create an over-complete library of beats, the MIT-BIH [12] database of EKG signals was used. Utilizing the provided beat annotations, existing EKG signals were split into 300 sample chunks centered on the R-Wave which could be represented as vectors such as $X$ from Fig. 5. These vectors are then combined into a matrix $(B)$ creating the dictionary of EKG beats. The singular value decomposition could then be performed on this matrix, decomposing the beats into a complex unitary matrix $U$, a diagonal matrix $\Sigma$, and a complex unitary matrix $V^T$ such that $B = U\Sigma V^T$. The columns of $U$ represent the eigen-beats or modes, hierarchically arranged based on their ability to describe the variance in the beat library. The eigen-beats

provide a basis from which we can represent the original beats. The singular value decay is shown in Fig. 4 where the relative importance of a given eigen-beat is plotted against the mode number.

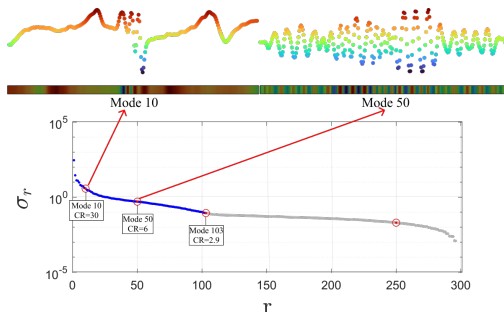

Fig. 4. Singular value decay of an over-complete library of heart beats.

The columns of $U$ from the SVD of the over-complete beat dictionary are selected to form the tailored basis $Psi_r$. The number of columns selected from $U$ to build $Psi_r$ can be empirically chosen by the optimal modal truncation value [13] or by using the Hammersley-Chapman-Robbins (HCR) bound [14]. In this paper the number of columns selected is instead determined by a varying CR such that performance of our method can be compared against other methods as the CR increases.

### C. Sensing Matrix Determination

A study performed on electrophysiologists tracking their eyesight while analyzing an EKG signal showed that less than 20% of the signal is actually viewed prior to classification of the beat [15]. It therefore comes as no surprise that particular portions of the signal are more significant than others which is translated into particular sampling locations of the signal. Similarly, optimal sampling locations of a tailored basis $\Psi_r$ have been shown to exist such that a measurement matrix $C$ can be built that minimizes the condition number of $\Theta$. These optimal sampling locations work to maximize the signal reconstruction accuracy while also increasing the CR.

Greedy optimization methods such as QR pivoting have been shown to expose optimal sampling locations of a tailored basis $\Psi_r$. The method proposed in this paper utilizes QR pivoting to extract a column permutation matrix $C^T$ containing the pivot locations. The first $M$ pivot locations are used to construct a measurement matrix $C$ where $M$ is chosen based on the CR ($M = N/CR$). Fig. 5 depicts the pivot locations in red against an uncompressed beat $x$ which is sampled by $C$ to create a compressed measurement $y$.

Not only do the pivot locations used to form $C$ determine the locations of $x$ which are sampled, but they also determine the rows of the tailored basis $\Psi_r$ which are activated when $\Theta$ is created. This is demonstrated pictorially in Fig. 6 where (left) depicts the tailored basis as a matrix, with rows activated by the QR pivot location and (middle & right) shows the tailored basis plotted as a surface map with a single beat $x$ and its sample locations $C$ translated into the compressed vector $y$.

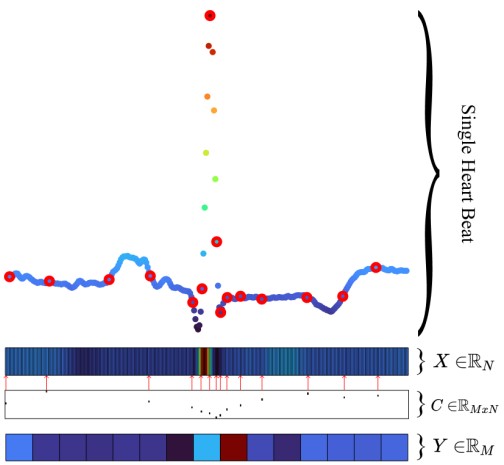

Single Heart Beat

$\} X \in \mathbb{R}_N$

$\} C \in \mathbb{R}_{MxN}$

$\} Y \in \mathbb{R}_M$

Fig. 5. Subsampled EKG signal using pivot locations from QR pivoting.

## D. Signal Reconstruction

A tailored basis $\Psi_r$ is generated in advance based on an over-complete library of beats from the device. The QR factorization is performed on this basis to determine the pivot locations which are then used to build a measurement matrix $C$. The measurement matrix is communicated to the device such that the optimal sampling locations are known for a given beat. The device need not know of the tailored basis nor perform any factorization to determine sampling locations. As the EKG signal is recorded the hardware will provide beat detections. These beats will then be used to segment the signal into compressible, R-Wave centered, windows which will be sampled via the measurement matrix. These compressed measurements will be stored on the device and eventually transmitted off the device to a bed-side monitor or a mobile application.

Once offloaded the compressed samples can easily be reconstructed into high-dimensional EKG signals by multiplying each sampling window by the pseudo-inverse of $\Theta$ to determine a low-rank vector of coefficients $a$ which can then be represented in the tailored basis $\Psi_r$ as a reconstructed signal $\hat{x}$. Because $\Theta$ is a square matrix, the pseudo-inverse can be calculated in constant time which scales linearly with the number of non-zero entries of $C$ determined by the compression ratio. This eliminates the need for the greedy, iterative convex solving algorithms needed to solve for $\hat{s}$ in traditional compressed sensing.

## IV. Performance Evaluation

To evaluate the method proposed in this paper against other state of the art EKG CS methods, an experiment was designed to evaluate signal reconstruction accuracy, on-device computational burden and reconstruction time of each method as it scaled with increased compression ratios. The compression ratio (CR) is calculated from the number of compressed measurements $M$ from the un-compressed measurements $N$ as in Eq. 4.

$$CR = \frac{N}{M} \qquad (4)$$

Real EKG signal data was used from the MIT-BIH arrhythmia database [12] which not only contain a wide range of patients, but also include common arrhythmias which an be particularly difficult to characterize. The database used was comprised of 47 patients who were sampled for 30 minutes at a sample rate of 360Hz with 11-bit resolution.

A number of existing CS methods were adapted from literature for comparison in this paper based on their reported performance. DDBD measurement matrices paired with a learned dictionary as a transform basis [6] referred to in this paper as DDBD-Dict was shown to be the highest performing of the existing CS methods. The wavelet transform has also been shown to be an excellent transform basis; Specifically the RBIO 5.5 transform [7] is used in conjunction with the DDBD measurement matrix as a CS method of comparison, referred to as DDBD-Wavelet. Other sampling matrices such as the Bernoulli binary matrix [8] and the Gaussian normal measurement matrix [9] were used in conjunction with a DCT transform basis [4] referenced in this paper is Bernoulli-DCT and Gaussian-DCT, respectively.

All of the CS methods compared in this paper require convex optimization to solve for $\hat{s}$. OMP, BP and IRLS were all tested against these methods with BP performing the best for both reconstruction accuracy and reconstruction time, confirming findings from other papers [7].

## A. Experimental Design

To evaluate the method proposed in this paper, all heart beats were extracted for the entire 30 minute EKG segment from each patient in the MIT-BIH arrhythmia database resulting in over 700,000 compressible windows to test against. 10,000 random sub-samples of these beats were taken to eliminate bias towards an individual patient or arrhythmia. Each of these compressible beat windows was then compressed and reconstructed using the method of choice and evaluated for signal reconstruction accuracy, computational burden on the device and time to reconstruct the signal. An example of a reconstructed EKG signal along with its corresponding error utilizing our tailored sensing method, the DDBD-Dict method and Bernoulli-DCT method (abbreviated Bern-DCT) are shown in Fig. 7.

## B. Reconstruction Accuracy

The reconstruction accuracy of a given beat was evaluated based on the percent root mean square difference (PRD) and the signal to noise ration (SNR) calculated in Eq. 5 and Eq. 6 where $x(n)$ is the original signal, $\hat{x}(n)$ is the reconstructed signal and $\bar{x}(n)$ is the mean signal.

$$PRD = \sqrt{\frac{\sum_{n=1}^{N}(x(n) - \hat{x}(n))^2}{\sum_{n=1}^{N} x(n)^2}} \qquad (5)$$

$$SNR = 10\log\left(\frac{\sum_{n=1}^{N}(x(n) - \bar{x}(n))^2}{\sum_{n=1}^{N}(x(n) - \hat{x}(n))^2}\right) \qquad (6)$$

PRD is often referred to as the gold standard of measuring EKG reconstruction accuracy because of its ability to capture

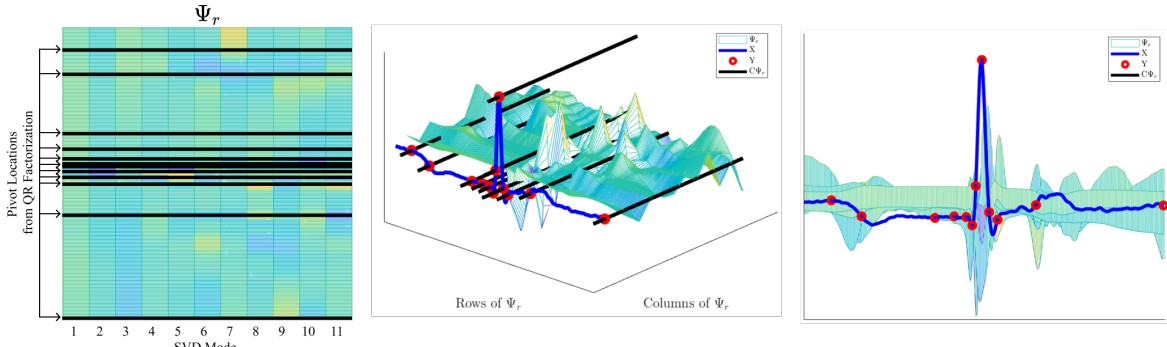

Fig. 6. QR pivot locations ($C$) plotted against the tailored basis ($\Psi_r$). (left) $\Psi_r$ as a matrix with rows selected by $C$. (middle/right) Single beat ($x$) with sample locations ($y$) plotted against the $\Psi_r$ as a surface mesh.

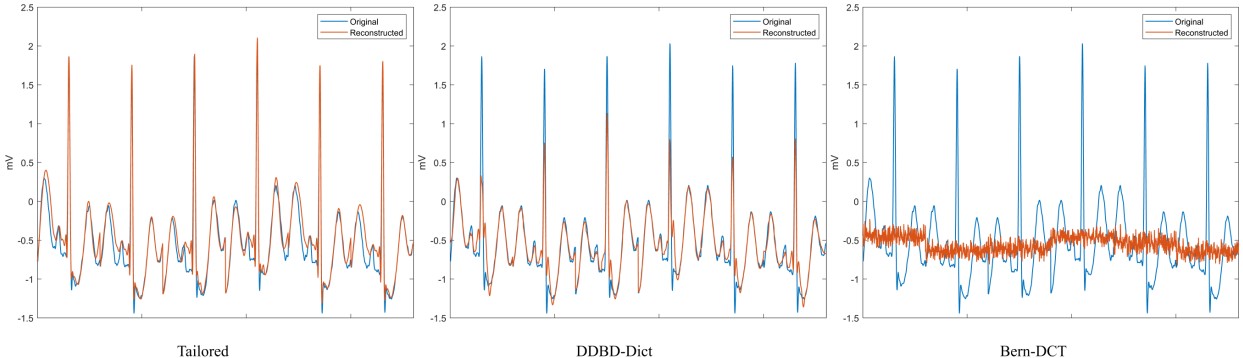

Fig. 7. EKG signal reconstruction comparison. (top) Reconstructed vs original signal. MIT-BIH patient 228 with a CR of 20.

beat characteristics as well as significant amplitude errors. A PRD between 2-9% has been shown to be sufficient for both visual and algorithmic analysis [16]. The average PRD for each method is plotted against a varied CR in Fig. 8 which shows that the dictionary based transform bases both significantly out perform all other CS methods analyzed in this paper.

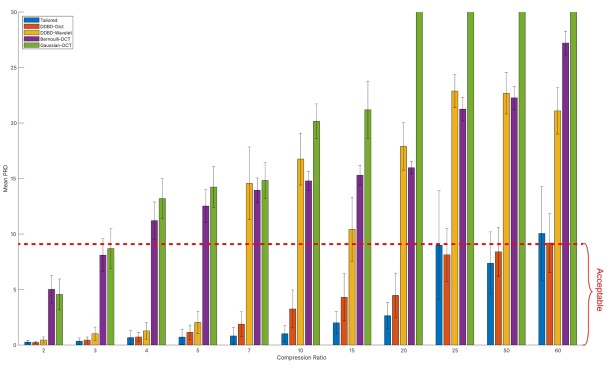

Fig. 8. Mean PRD.

To better visualize the difference between the TS method and the DDBD-Dict CS method, the mean PRD for each was isolated and plotted in Fig. 9. The TS method out-performs DDBD-Dict for CRs below 50 At higher compression ratios the mean PRD appears similar, but the standard deviation indicates poor performance for both TS and DDBD-Dict.

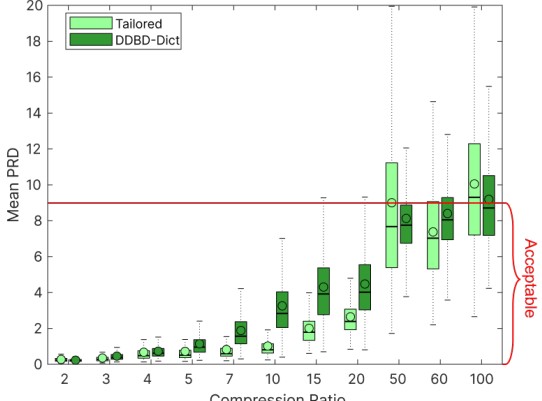

Fig. 9. Mean PRD for the Tailored and DDBD-Dict CS methods.

While a low PRD is always desired for EKG signal reconstruction, this figure does not capture noise in the reconstructed signal which can lead to poor performance in algorithm based EKG analysis. A SNR of 20 dB or higher has been shown to be sufficient for arrhythmia classification based algorithm analysis of EKG signals [16]. The mean SNR for each CS method is plotted against a varying CR in Fig. 10.

The SNR for the TS method proposed in this paper significantly out-performs other CS methods across increasingly large CRs. Our methods achieve an average of 8% higher SNR across all CRs tested.

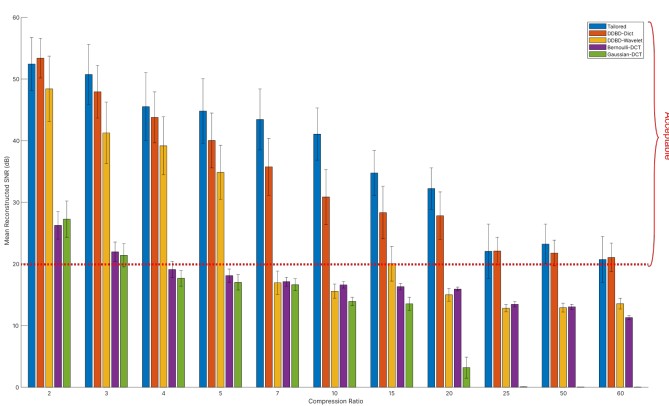

Fig. 10. Mean SNR.

### C. On-Device Sensing Burden

Two extremely important features of ICM devices is size and longevity meaning efficiency is critical for these devices. An important aspect of battery drain for any modern micro-controller application is the ability for the device to enter a periodic sleep mode with an extremely low clock rate. Cardiac based micro-controller devices will often enter a sleep mode in-between heart beats, thus it is critically important that the amount of computation performed on a per-beat basis (such as compression) is as small as possible.

To assess the computational burden imposed on a device by a specific CS method, the battery drain was evaluated by measuring both the sampling power and transmission power. The sampling power ($PWR_{smp}$) reflects the number of addition and multiplication operations required to apply the measurement matrix $C$ to a compressible EKG data window. Transmission power ($PWR_{tx}$) was calculated based on the energy needed for Bluetooth® Low Energy (BLE) transmission, measured on a per-bit basis. The power consumption in this study was based on the Microchip PIC32MX470F512L microcontroller, compiled with XC32 on a MIPS32 instruction set, and a Texas Instruments CC2540 BLE transceiver. The PIC32MX470F512L features an addition time ($t_a$) of 60 ns, a multiplication time ($t_m$) of 90 ns, an active power draw ($P_{awake}$) of 0.198 W, and a sleep power draw ($P_{asleep}$) of 0.165 mW. The CC2540 BLE transceiver has a transmission power of 84 mW and a transmission rate of 8 uS per bit, resulting in a power-per-bit transmission cost ($P_{ble}$) of 84 nJ.

$PWR_{smp}$ was calculated using Eq. 8 where $a$ represents the number of additions and $m$ represents the number of multiplications required to apply the measurement matrix $C$ to the uncompressed signal $x$. While $PWR_{tx}$ is calculated in Eq. 9 where $r$ represents the resolution of the signal at a nominal 12 bit resolution. The total power ($P_t$) is the summation of the sampling and transmission power ($PWR_{total} = PWR_{smp} + PWR_{tx}$).

$$t_{smp} = ((a \times t_a) + (m \times t_m)) \qquad [sec] \qquad (7)$$

$$PWR_{smp} = (t_{smp} \times P_{awake}) - (t_{smp} \times P_{asleep}) \qquad [J] \qquad (8)$$

$$PWR_{tx} = (M \times r) \times P_{ble} \qquad [J] \qquad (9)$$

A plot of the sampling power from each CS method is showing in Fig. 11 where clearly the proposed method of tailored sampling has significantly better power drain characteristics. This is because our method does not require any addition or multiplication instructions to sample since each row of $C$ only has a single pivot location. This implies that the only power drain comes from the transmission of the signal and the indexing of the beat. Since each method will have the same $PWR_{tx}$ for a given CR, it does not add anything of value to the plot. The DDBD measurement matrix methods have a constant sampling power draw since the number of samples per row of $C$ increases as the CR increases meaning a constant number of additions are performed.

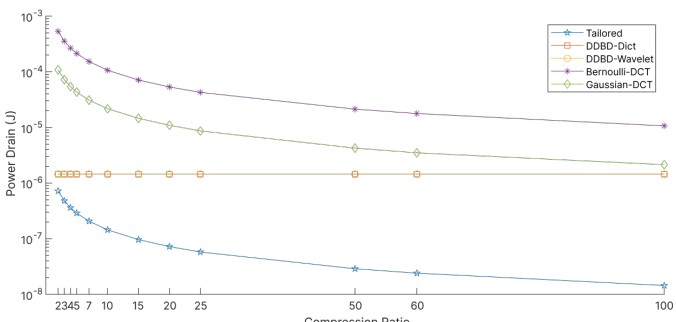

Fig. 11. Sampling power draw logarithmic plot

### D. Reconstruction Time

The time to reconstruct a compressed EKG sample into a full EKG signal is of critical importance in the world of WCM and ICM devices as the scale of data being offloaded is massive and any additional server side compute can be costly. The method proposed in this paper drastically reduces the reconstruction for a given compressible segment as TS does not require a costly, iterative, convex optimization solver to reconstruct the signal. Instead a compressed sample $y$ can be reconstructed directly in constant-time by multiplying by the inverted matrix $\Theta^+$ and then multiplying by the tailored basis $Psi_r$.

Signal reconstruction time was measured for each CS method and for every beat compressed. This reconstruction was performed on an HP ZBook Studio G5 which has an Intel i9-8950HK processor, 32 GB of RAM and a NIVIDIA Quadro P1000 GPU. The results in Fig. 12 show how the proposed method outperforms all other methods in reconstruction time with an average of 93% lower reconstruction time across all CR's.

## V. CONCLUSION

### A. Summary of Findings

In this paper a novel compressed sensing method is proposed for real-time EKG acquisition on implanted medical devices. Our method employs a novel signal segmentation scheme which utilizes existing, on-device beat detections to segment an EKG signal into beat-based compressible windows. Using a tailored transform basis generated from the

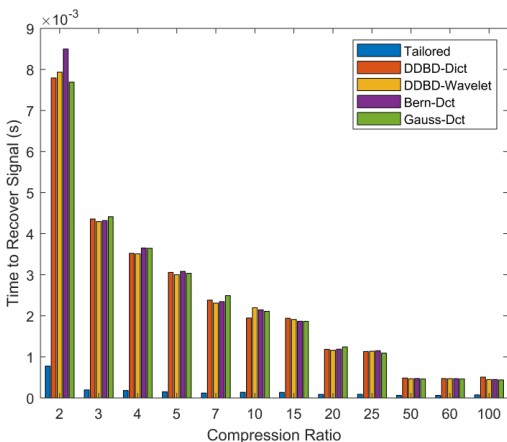

Fig. 12.  Mean signal reconstruction time

decomposition of an over-complete library of known beats, our method selects optimal sampling locations determined from the QR factorization of the basis to compress each beat-based window for storage and later transmission. Finally, after transmission, our method is able to directly solve for the reconstructed signal without the need for a convex optimization solver.

The TS method proposed in this paper boasts better signal reconstruction without any on-device computational burden and a 93% reduction in signal reconstruction time. Additionally, the signal segmentation proposed on this paper allows for previously inconceivable compression ratios. The findings in this paper offer the possibility for a new type of implanted medical device which can bypass the necessity to develop expensive, non-updatable, on-device detection algorithms and instead focus on a sense-store-transmit based device that leaves the computation to compute-capable hardware. In addition, the findings in this paper offer an exciting future for edge-based sensor data acquisition where a limited knowledge of a signal can be used to significantly compress the signal.

### B. Limitations and Future Work

While the findings in this paper offer exciting results for the compression of EKG signals, there exist a number of limitations and some exciting future work bulleted below.

**Limitations:**

- Beat-based signal segmentation has a required size to cohere to the tailored basis. Extremely high heart rates would likely need a separate tailored basis to capture their characteristics.
- The PRD values calculated are biased by our method since we center on the exact peak location thus never miss the peak amplitude of the signal.
- The tailored basis created may require adjustments based on heart signal characteristics sensed from the location of implant.

**Future Work:**

- A per-patient, or auto-adjusting tailored basis would be interesting to experiment with although regulatory bodies

are hesitant to endorse non human-in-the-loop algorithm updates.

- A tailored basis for different beat types would be interesting as well, but requires on-device beat classification.
- The implementation of this method on real hardware and analyzed in a noisy environment would be interesting to test robustness.

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
