# OpenReview forum: "Efficient Compressed Sensing for Real-Time Electrocardiogram Acquisition on Low-Power Implanted Medical Devices"
_IEEE.org/EMBS/BHI/2024/Conference — IEEE BHI'24_

### Official Review · Reviewer_Ct3B · 2024-08-02
**Good and novel approach, results need to be reorganized and better presented**

**Overall Rating:** 7
**Confidence:** 2

**Other Quality Metrics:**

(a) Clarity of writing: GOOD
(b) Clinical Significance: GOOD
(c) Methodological Novelty FAIR, to the best of my knowledge
(d) Experiments and Results: FAIR

**Questions For The Authors:**

•	If you did not perform cross-validation, please do it and describe the procedure
•	Please, state clearly which approaches are you comparing yourself with, and according to which KPIs, in a more concise and schematic way. Then, stick to this logic in the entire Results section. For example:
o	Why do Figures 8 and 10 compare six different approaches while Figure 9 compares only two?
•	Clarify whether higher or lower values are preferred for the different KPIs (PRD, SNR). Although it may seem trivial, this helps with immediate comprehension. For example:
o	Which area of Fig. 8 is optimal?
•	Please add a statistical test to compare the boxplots in Figure 9 to determine if the performance differences are significant.
•	For Figure 8, please specify what each point represents and indicate which areas of the graph denote optimal performance. Add a caption and increase the text size.
•	Figure 9 suffers from the same issues as Figure 8 and is also unreadable due to its size.
•	Are the lines in Figures 8 and 9 averages? If not, was cross-validation performed? If so, could the standard deviation area be shown? Please add both cross-validation and standar-deviation ribbons around the lines.
•	The issues highlighted in Figures 8 and 9 are also present in Figure 11.
•	The CR seems to be compared with other KPIs (Figures 8, 9, 11, 12), but different formats are used each time, with varying graphic details. Could the choices be standardized? The use of boxplots and barplots seems to underlie a multiplicity of measures per CR value, which is however not present when comparing CR with PRD and SNR though.
•	Please describe any limitations and potential future work in the conclusions.
•	The 93% value is cited multiple times in the paper, but never demonstrated in the results. Please, highlight where that value comes from, and the other values as well
•	The work would benefit of the presence of Tables describing the results, on the contrary the Microchip characteristics could be in the text and the relative tables in the Supplementary Materials
MINOR
•	Avoid including specific names such as "PIC32MX470F512L" and "CC2540 BLE" in table titles. Consider using "Technical specifications of the employed Microchip..." or a similar alternative.
•	Increase the size of the figures and remove Table IV, which could be replaced with plain text. The figures are very difficult to read.
•	The first sentence introducing the two acronyms is somewhat confusing.
•	The organization of Figure 6 is unclear. Please rephrase to improve clarity. If the first phrase in a caption is the title, please make it bold.

**Strengths:**

- The relevance of the topic and objective is well addressed, as evidenced by the proposed results.
- The method has the potential to be applied to other fields (e.g., edge computing with different types of sensors, as highlighted by the authors).
- The paper is well-written, with information organized in a rational, clear, and concise manner (for what concerns the writing part)

**Summary Of The Paper:**

The paper introduces a novel approach to compressed sensing for use in implanted devices. To achieve this, the authors utilize prior knowledge to enhance both storage efficiency and real-time analysis of ECGs.

**Weaknesses:**

- The validation section is somewhat lacking: cross-validation should be added, detailed information on the validation dataset should be provided, and the presentation of results needs improvement in both form and substance.

- Comparisons with other methods are disorganized and unclear, with an overuse of acronyms making it challenging to navigate the different approaches and their specific features.

- While many techniques are introduced and compared, which is valuable, the proposed methodology should be highlighted more clearly to allow readers to immediately grasp its performance.

- The figures and their captions are not very clear: some are too small and surrounded by excessive white space, others are inadequately introduced and explained, and some lack proper captions.

- The paper contains some typographical errors.

---

### Official Review · Reviewer_kCBF · 2024-08-17
**Review of Efficient Compressed Sensing for Real-Time Electrocardiogram Acquisition on Low-Power Implanted Medical Devices**

**Overall Rating:** 8
**Confidence:** 2

**Other Quality Metrics:**

- Clarity of writing: Excellent
- Clinical Significance: Excellent
- Methodological Novelty: Excellent
- Experiments and Results: Ecellent

**Questions For The Authors:**

- How does the proposed method handle patient-specific variability in EKG signals? Are there any potential challenges in adapting the TS method across different patient populations?
- The experiments were conducted using data from the MIT-BIH arrhythmia database. Have you considered validating your method on other datasets or in clinical settings to further support your claims?

**Strengths:**

### Clarity and Structure
- The paper is very well organized and generally easy to read. It has a clear and logical flow, with sections that are well-defined and easy to follow.
- The background section and related work section clearly cover technical details on compressed sensing, and a table of notation is provided, which helps the reader to follow.
### Motivation of the Work
- This paper is well motivated with a properly defined problem, establishing the need for more efficient EKG data compression methods in implanted medical devices. The authors effectively highlight the limitations of existing approaches and justify the need for their proposed solution.
### Presentation
- Besides the table mentioned earlier, the author provided a sufficient number of figures, which are well-constructed and enhance the understanding of the concepts presented. The figures are appropriately labeled and referenced throughout the text.
### Technical Novelty
- The paper presents a novel approach by introducing Tailored Sensing (TS) for EKG data acquisition, which optimizes compression and reconstruction by utilizing prior knowledge of the signal. This approach is innovative, particularly in the context of low-power, implanted medical devices, and it offers a significant improvement over existing methods.
### Experiment Design
- The experimental design is thorough. The use of real EKG data from the MIT-BIH arrhythmia database enhances the validity of the findings. Moreover, the authors provide sufficient experimental details, which can help their work to be replicated. Additionally, the experiments benchmark relevant methods. The performance improvment is significant, especially in the reconstruction time.
### Writing
- The writing is generally clear and free of major errors.

**Summary Of The Paper:**

This paper introduces a method called Tailored Sensing (TS) to enhance EKG data acquisition in implantable devices. This method optimizes signal compression and reconstruction by leveraging prior knowledge of EKG signals, significantly reducing on-device computational demands and achieving high compression ratios without compromising signal quality. By author's claim, the TS approach also reduces signal reconstruction time by 93%, making it well-suited for low-power, long-lasting medical devices that require efficient data storage and transmission for external analysis.

**Weaknesses:**

### Lack of discussions on limitations and future works
One notable area where the paper could be improved is in its discussion of limitations and future directions. While the proposed Tailored Sensing (TS) method is innovative and effective, the paper does not fully address potential limitations or challenges that could arise when implementing this approach in real-world scenarios.

### Some typos:
- Section III.B: "decompisition" should be "decomposition"
- Section IV: "exiting" should be "exciting"​

---

### Official Review · Reviewer_W77r · 2024-08-18
**Efficient Compressed Sensing for Real-Time Electrocardiogram Acquisition on Low-Power Implanted Medical Devices**

**Overall Rating:** 6
**Confidence:** 4

**Other Quality Metrics:**

(a) Clarity of writing; - Fair
 (b) Clinical Significance; - Good
(c) Methodological Novelty; - Good
(d) Experiments and Results -Good

**Questions For The Authors:**

- Any discussion on why this method outperforms existing CS approach?
- Could you provide more details on the experiment setup?
- Any in-depth analysis on the computational cost, in comparison with other approaches?

**Strengths:**

- The overall flow has novelty in reconstructing signal from a sub-sampled data
-The proposed flow is particularly a fit for implanted wearable device where there is huge need to reconstruct sub-sampled data in a time efficient manner.
-The experiments is validated among 47 patients.

**Summary Of The Paper:**

This paper proposed a compressed sensing approach to increase the efficiency of data acquisition for EKG signal on low-power implanted medical device.  Overall, this paper addressed an important issue in implanted medical device. The newly proposed CS approach has novelty in technical side. However, there methed is not clearly detailed and experimental setup needs more information to justify the superiority of proposed method.

**Weaknesses:**

- The approach to segment signal in Section 3A is very arbitrary and may suffers from noise.
- More details on building dictionary in the experiment is expected.
- The comparison methods are not consistency among figures regarding quantitative results. For example, wavelet results are shown in Fig. 7 but not other figures. It is important to show the quantitative results rather than only qualitative results in Fig. 7.
-In Fig. 10, if the compression ratio is 100, the reconstructed signal should equal to original signal, based on Eq. 12, the SNR should be 0.
-What do the different colors mean in Fig. 3?
- What is the exact portion of the runtime for algorithm needed in this implementation? It is noted that even for 100% compressed ratio, there is a computational cost for the runtime in Fig. 12.

---

### Decision · Program_Chairs · 2024-09-23

Accept